# Using machine learning to determine the nationalities of the fastest 100-mile ultra-marathoners and identify top racing events

Beat Knechtle[1,2]*, Katja Weiss[2], David Valero[3], Elias Villiger[2], Pantelis T. Nikolaidis[4], Marilia Santos Andrade[5], Volker Scheer[3], Ivan Cuk[6], Robert Gajda[7], Mabliny Thuany[8]

1 Medbase St. Gallen Am Vadianplatz, St. Gallen, Switzerland, 2 Institute of Primary Care, University of Zurich, Zurich, Switzerland, 3 Ultra Sports Science Foundation, Pierre-Benite, France, 4 School of Health and Caring Sciences, University of West Attica, Athens, Greece, 5 Department of Physiology, Federal University of São Paulo, São Paulo, Brazil, 6 Faculty of Sport and Physical Education, University of Belgrade, Belgrade, Serbia, 7 Center for Sports Cardiology at the Gajda-Med Medical Center in Pułtusk, Pułtusk, Poland, 8 Faculty of Sports, University of Porto, Porto, Portugal

* beat.knechtle@hispeed.ch

**Data Availability Statement:** Availability of Data and Materials For this study, we have included official results and split times from the DUV Ultra-Marathon Statistik (statistik.d-u-v.org/geteventlist.

## Abstract

The present study intended to determine the nationality of the fastest 100-mile ultra-marathoners and the country/events where the fastest 100-mile races are held. A machine learning model based on the XG Boost algorithm was built to predict the running speed from the athlete's age (*Age group*), gender (*Gender*), country of origin (*Athlete country*) and where the race occurred (*Event country*). Model explainability tools were then used to investigate how each independent variable influenced the predicted running speed. A total of 172,110 race records from 65,392 unique runners from 68 different countries participating in races held in 44 different countries were used for analyses. The model rates *Event country* (0.53) as the most important predictor (based on data entropy reduction), followed by *Athlete country* (0.21), *Age group* (0.14), and *Gender* (0.13). In terms of participation, the United States leads by far, followed by Great Britain, Canada, South Africa, and Japan, in both athlete and event counts. The fastest 100-mile races are held in Romania, Israel, Switzerland, Finland, Russia, the Netherlands, France, Denmark, Czechia, and Taiwan. The fastest athletes come mostly from Eastern European countries (Lithuania, Latvia, Ukraine, Finland, Russia, Hungary, Slovakia) and also Israel. In contrast, the slowest athletes come from Asian countries like China, Thailand, Vietnam, Indonesia, Malaysia, and Brunei. The difference among male and female predictions is relatively small at about 0.25 km/h. The fastest age group is 25–29 years, but the average speeds of groups 20–24 and 30–34 years are close. Participation, however, peaks for the age group 40–44 years. The model predicts the event location (country of event) as the most important predictor for a fast 100-mile race time. The fastest race courses were occurred in Romania, Israel, Switzerland, Finland, Russia, the Netherlands, France, Denmark, Czechia, and Taiwan. Athletes and coaches can use these findings for their race preparation to find the most appropriate racecourse for a fast 100-mile race time.

php). The datasets used and/or analyzed during the current study are available from the corresponding author on reasonable request.

**Funding:** The author(s) received no specific funding for this work.

**Competing interests:** The authors have declared that no competing interests exist.

## Introduction

Ultra-endurance events lasting more than 6 hours include ultramarathon running races [1]. The popularity of these events has increased significantly in the last 25 years, particularly in ultra-marathon races, where an exponential increase in participation has been observed [2, 3]. The ultra-marathon race distance of 100 miles (161 km) is highly popular, especially in the United States [4]. The high popularity of the 100-mile race distance among ultra-marathoners has led to a high level of scientific interest among researchers [5–9]. The main topics of scientific interest have included fluid and electrolyte metabolism, the heart structure and function of the 100-mile ultra-marathoner, successful race performance, pacing, nutrition, anthropometry, age, mental toughness, sleep, muscle damage, skeletal and renal health, overuse injuries, metabolomics, and pain perception [5–7]. A large number of studies have been performed about the 'Western States Endurance Run' [8, 9], where the first paper appeared in 1987 [6].

The largest body of research has been performed on fluid and electrolyte metabolism [10], focusing on specific aspects such as exercise-associated hyponatremia [11, 12] and fluid metabolism [13, 14]. Considering the heart of the 100-mile ultra-marathoners, aspects such as cardiac adaptation [15], the heart rate variability [5], alterations in cardiac mechanics [16], and the right ventricle [17] were of scientific interest. Other aspects of 100-mile ultra-marathoners such as the age of the best performance [18], the sex difference in performance [19], age group performances [18], physiological aspects [20], nutrition [21], mental toughness [7], pacing [22], and the use of non-steroidal anti-inflammatory drugs [23, 24] have also been studied extensively.

Since most of the research on 100-mile ultra-marathoners has been performed in races held in the United States, most 100-mile runners originate from that country. However, a study conducted to understand the best countries competing at 100 miles, using a macro-to-micro analysis, showed that most of the athletes were from the American and European continents, despite the observation of the fastest being from Africa [25]. An analysis of each continent showed that women from Sweden, Hungary and Russia presented the best performances in the top three, top 10 and top 100, while the fastest men were from Brazil, Russia and Lithuania [25]. However, we do not know (i) athletes of which nationality are the fastest in 100-mile ultra-marathon running and (ii) where the fastest 100-mile race courses are located worldwide. In this context, we undertook this research to determine the country of origin of the fastest runners and the location of the fastest race courses. These insights would help athletes and coaches to better plan their race strategy to obtain a fast race time. We hypothesized, based on these recent findings, that our study would confirm recent findings that most of the 100-mile runners would originate from Europe and America, that the fastest 100-mile runners would be found to originate from Africa, and that the fastest race courses would be situated in Africa, Europe and/or America.

## Methods

### Ethical approval

This study was approved by the Institutional Review Board of Kanton St. Gallen, Switzerland, with a waiver of the requirement for informed consent of the participants as the study involved the analysis of publicly available data (EKSG 01/06/2010). The study was conducted following recognized ethical standards according to the Declaration of Helsinki adopted in 1964 and revised in 2013.

## Data set and data preparation

The race data was obtained from DUV Ultra-Marathon Statistik (statistik.d-u-v.org/getevent-list.php) by the end of 2022. The data were accessed July 11, 2023, for research purposes. The raw 100-mile sample contained 172,394 race records, with the United States accounting for around 70% of the sample, while there were also numerous countries with just one or two race records. Each race record included the athlete's name, age group, gender and country of origin, the race location and year, the race distance, and the athlete's race time, from which the race speed was calculated. ISO3 codes were used for the country information. After discarding any incomplete or incorrect instances and filtering out countries with a very low number of records, a total of 172,110 race records from 65,392 unique runners from 68 different countries participating in races held in 44 different countries were used for analyses. To minimize the potential effect of outliers, a minimum of 10 race records was set per country to qualify for the analysis.

## Statistical analysis

First, two independent ranking tables were created, aggregating the race records by country of origin and event, and then sorting each list of countries by number of race records. To reduce noise and ensure that the results were statistically representative, race records from athlete countries with less than 15 records or less than five unique runners were removed, and race records from event countries with less than 10 records were removed. Descriptive statistical results for each country are summarized in the ranking tables, where the tables index also serves as a key to the Partial Dependence Plots (PDP). We then built and evaluated a non-linear machine learning (ML) predictive regression model and looked into the model logic through some explainability tools. The algorithm used for building the model is the popular XG Boost. XG Boost (xgboost.readthedocs.io/en/stable/) belongs to the family of gradient-boosting tree-ensemble algorithms and is widely used to solve classification and regression problems in data science.

**XG Boost regression model.** The model was designed to use the following variables as predictors or inputs to the model: "Athlete_gender_ID", "Age_group_ID", "Athlete_country_ID", and "Event_country_ID". The predicted variable, or algorithm output, was the Race (running) speed (km/h). Before the data could be fit in the model, the predictors had to be numerically encoded, Th. Athlete gender variable was encoded as female = 0 and male = 1. The Age group variable was already numerically encoded in 5-year age groups (except group 18, which represents runners of less than 20 years, and group 75, which represents 75 years and older). The Athlete country and Event country variables were encoded based on their position in the respective ranking tables. Fig 1 illustrates the setup, with the variables used as predictors or inputs and the race (running) speed prediction as the model output.

**Model training and evaluation strategy.** A hold-out evaluation strategy was used to train and evaluate the model, executing a simulation with different test splits and combinations of several estimators and learning rates. Two evaluation metrics, MAE (Mean Absolute Error) and $R^2$, were calculated. Also, the model relative features importances, partial dependence plots (PDP) and prediction distribution plots were calculated and are displayed in the results section. In addition to the model interpretability analysis, a set of descriptive target plot charts show the predictor values, group sizes, and the group's average speed, helping to set expectations for the PDP and prediction charts.

After several iterations and tests, the optimal model parameters and accuracy scores were:

- 500 estimators (learners or trees)

**Fig 1. XG Boost model.**

- Learning rate of 0.5
- $R^2$ score of 0.23 (in-sample test)
- MAE of 0.87 km/h

**Model interpretation.**   The 'optimal' model accuracy score of $R^2$ = 0.23 indicates an existing but moderately weak effect of the predicting variables in the model output. To assess how each predictor contributed to the model output, we computed the importance of the model's relative features, the PDP plots, and the model prediction distributions. The PDP plots show the relative amount of change on the model output for each predicting variable's different values with respect to a reference value (value 0). The prediction distribution plots use boxplots to show the distribution of the model predictions of average race speed. Descriptive statistical values are given in terms of frequencies (counts), mean, standard deviation (std), minimum values (min), and maximum values (max), and also with median values (in the box plots). All computation and analysis were done using a Jupyter Notebook (Google Colab) and Python and associated libraries (pandas, numpy, xgboost, pdpbox, sklearn, matplotlib, sns).

## Results

The qualifying sample used for analysis consists of 172,110 race records from 65,392 unique runners from 68 different countries participating in races held in 44 different countries. Table 1 presents the country rankings by number of race records and unique runners. The mean race speed is color-coded, with darker colors corresponding to higher values (faster running speed). The first column in the ranking tables is the index to interpret the PDP charts. The United States accounted for the highest participation in both athlete country and event country rankings, followed by Great Britain, Canada, South Africa, Japan, Germany, and Australia.

### Event country ranking

The country of event ranking table, with 42 countries, is shown in Table 2. Most runners competed in races held in the United States, Great Britain, South Africa, Japan, and Germany.

**Table 1. Athlete country ranking table.**

| | Athlete country | Race speed (km/h) | | | | Race records | Unique runners |
|---|---|---|---|---|---|---|---|
| | | mean | std | min | max | | |
| 0 | USA | 6.053 | 1.147 | 1.176 | 15.409 | 117583 | 37833 |
| 1 | GBR | 6.428 | 1.479 | 2.222 | 13.977 | 11193 | 5501 |
| 2 | CAN | 6.211 | 1.127 | 2.572 | 16.974 | 5553 | 2080 |
| 3 | RSA | 7.314 | 1.349 | 2.410 | 13.469 | 5547 | 2657 |
| 4 | JPN | 5.349 | 1.348 | 2.443 | 12.428 | 5084 | 2913 |
| 5 | GER | 6.710 | 1.378 | 2.338 | 12.520 | 4439 | 2001 |
| 6 | AUS | 6.288 | 1.554 | 2.252 | 13.449 | 2894 | 1464 |
| 7 | SWE | 6.963 | 1.387 | 2.072 | 12.839 | 2094 | 895 |
| 8 | CHN | 4.381 | 1.155 | 2.715 | 12.087 | 1700 | 1491 |
| 9 | ITA | 6.467 | 1.891 | 2.700 | 12.363 | 964 | 568 |
| 10 | DEN | 7.296 | 1.325 | 3.379 | 11.851 | 958 | 493 |
| 11 | FRA | 6.462 | 1.946 | 2.403 | 12.013 | 953 | 644 |
| 12 | PHI | 4.989 | 0.972 | 3.025 | 9.984 | 919 | 384 |
| 13 | NZL | 6.082 | 1.527 | 3.382 | 12.389 | 896 | 526 |
| 14 | GRE | 5.121 | 1.344 | 3.696 | 13.665 | 808 | 425 |
| 15 | POL | 6.924 | 1.716 | 3.445 | 12.960 | 799 | 510 |
| 16 | BEL | 6.567 | 1.424 | 2.862 | 11.233 | 780 | 435 |
| 17 | TPE | 7.194 | 1.583 | 3.362 | 11.109 | 768 | 339 |
| 18 | NOR | 6.952 | 1.714 | 3.216 | 11.895 | 738 | 364 |
| 19 | NED | 6.807 | 1.361 | 3.338 | 11.777 | 728 | 369 |
| 20 | IRL | 6.913 | 1.518 | 2.735 | 11.667 | 606 | 325 |
| 21 | MEX | 6.242 | 1.487 | 3.136 | 12.600 | 558 | 239 |
| 22 | HUN | 7.746 | 1.563 | 1.277 | 12.745 | 503 | 289 |
| 23 | RUS | 7.717 | 1.636 | 3.597 | 14.034 | 424 | 294 |
| 24 | CZE | 7.136 | 1.850 | 2.440 | 12.905 | 376 | 217 |
| 25 | FIN | 7.697 | 1.355 | 4.014 | 11.666 | 346 | 169 |
| 26 | SUI | 6.584 | 1.532 | 2.797 | 11.819 | 331 | 160 |
| 27 | THA | 4.052 | 0.784 | 2.982 | 8.778 | 321 | 273 |
| 28 | ESP | 6.881 | 2.012 | 2.281 | 12.256 | 320 | 216 |
| 29 | KOR | 6.479 | 1.116 | 3.529 | 9.944 | 299 | 204 |
| 30 | ARG | 5.466 | 1.181 | 3.278 | 10.530 | 282 | 235 |
| 31 | MAS | 4.307 | 1.001 | 3.096 | 8.172 | 236 | 177 |
| 32 | HKG | 5.424 | 1.719 | 3.358 | 10.761 | 166 | 116 |
| 33 | AUT | 6.660 | 1.733 | 2.849 | 12.096 | 163 | 114 |
| 34 | IND | 6.207 | 1.330 | 3.741 | 11.178 | 156 | 101 |
| 35 | SGP | 5.307 | 1.047 | 3.186 | 9.297 | 149 | 104 |
| 36 | BRA | 6.496 | 1.902 | 3.428 | 11.654 | 110 | 65 |
| 37 | CRO | 5.882 | 2.067 | 3.355 | 10.937 | 103 | 88 |
| 38 | COL | 4.820 | 1.535 | 3.183 | 9.237 | 103 | 66 |
| 39 | SLO | 6.310 | 2.162 | 3.355 | 11.785 | 90 | 67 |
| 40 | SVK | 7.488 | 1.797 | 3.298 | 12.184 | 80 | 42 |
| 41 | VIE | 4.209 | 0.656 | 3.612 | 6.902 | 66 | 41 |
| 42 | ISR | 7.353 | 1.543 | 4.010 | 10.487 | 66 | 66 |
| 43 | ROU | 6.908 | 2.153 | 3.209 | 13.411 | 62 | 42 |
| 44 | CHI | 5.700 | 0.949 | 2.779 | 8.677 | 62 | 39 |
| 45 | LAT | 7.626 | 1.796 | 4.478 | 11.462 | 62 | 36 |

*(Continued)*

**Table 1.** (Continued)

| | Athlete country | Race speed (km/h) | | | | Race records | Unique runners |
|---|---|---|---|---|---|---|---|
| | | mean | std | min | max | | |
| 46 | UKR | 7.900 | 1.974 | 4.738 | 12.831 | 59 | 32 |
| 47 | PAN | 5.877 | 1.058 | 4.139 | 9.461 | 58 | 17 |
| 48 | POR | 6.726 | 1.722 | 3.501 | 11.723 | 46 | 28 |
| 49 | SRB | 6.183 | 2.036 | 3.650 | 11.825 | 44 | 29 |
| 50 | PER | 5.854 | 1.351 | 3.250 | 9.649 | 43 | 17 |
| 51 | BUL | 6.749 | 2.046 | 4.196 | 12.568 | 37 | 22 |
| 52 | GUA | 6.332 | 1.251 | 4.480 | 9.551 | 37 | 12 |
| 53 | CRC | 6.010 | 1.095 | 3.392 | 8.613 | 36 | 16 |
| 54 | LTU | 8.738 | 2.352 | 4.311 | 14.365 | 35 | 21 |
| 55 | TUR | 6.647 | 1.782 | 3.781 | 9.976 | 30 | 14 |
| 56 | EST | 7.266 | 1.635 | 3.713 | 10.205 | 28 | 17 |
| 57 | BLR | 6.988 | 1.133 | 4.504 | 9.166 | 27 | 15 |
| 58 | VEN | 5.894 | 1.378 | 3.214 | 7.706 | 25 | 11 |
| 59 | PUR | 6.563 | 1.086 | 4.721 | 8.854 | 23 | 9 |
| 60 | IRI | 5.765 | 0.899 | 3.691 | 7.062 | 21 | 5 |
| 61 | INA | 4.492 | 0.592 | 3.473 | 5.539 | 20 | 12 |
| 62 | BRU | 3.978 | 0.679 | 3.203 | 5.437 | 19 | 12 |
| 63 | CYP | 5.156 | 0.793 | 3.860 | 6.399 | 18 | 11 |
| 64 | ISL | 5.664 | 0.981 | 4.180 | 8.039 | 18 | 9 |
| 65 | BIH | 6.621 | 1.754 | 4.276 | 9.889 | 16 | 9 |
| 66 | ECU | 6.305 | 1.573 | 3.416 | 9.223 | 16 | 9 |
| 67 | ZIM | 7.121 | 2.491 | 4.417 | 11.883 | 16 | 8 |

Std (standard deviation); min (minimum value); max (maximum value)

## Model features relative importances

The 'optimal' model can only explain 23% of the race speed variability through the four predictors at best, indicating that additional predicting variables should be added to the model in order to improve its accuracy. The model (Fig 2) rates *Event country* (0.49) as the most important predictor (based on data entropy reduction), followed by *Athlete country* (0.24), *Age group* (0.15), and *Gender* (0.13).

## Partial dependence plots (PDP)

The PDP plot shows the following: Model outputs are around 0.26 km/h higher for males than for females (Fig 3). The highest model outputs are given to runners in age groups 25–29 years and 30–34 years (Fig 4). Athlete country ID 54 (Lithuania) shows a distinct peak, matching the highest mean speed in the ranking table (Fig 5). Event country IDs 16, 19, 23 and 34 obtain the highest peaks in the corresponding PDP chart, although only 23 (Switzerland) and 34 (Romania) are among the fastest in the ranking table (Fig 6).

## Prediction distributions and target plots

The target plots represent a descriptive visualization of the 100 km race dataset by predictor and show the groups' sizes and average speeds. The prediction plots show the distribution of the XG Boost model output (the predicted race speed) by predictor value through a set of

**Table 2. List of event countries sorted by mean running speed.**

| Event country | | Race speed (km/h) | | | | Race records | Unique runners |
|---|---|---|---|---|---|---|---|
| | | mean | std | min | max | | |
| 0 | USA | 6.061 | 1.152 | 1.176 | 16.974 | 124323 | 40565 |
| 1 | GBR | 6.449 | 1.439 | 2.656 | 14.307 | 11343 | 6002 |
| 2 | RSA | 7.314 | 1.341 | 2.410 | 13.469 | 5581 | 2717 |
| 3 | JPN | 5.126 | 1.187 | 2.443 | 10.922 | 4397 | 2878 |
| 4 | GER | 6.813 | 1.251 | 2.836 | 12.600 | 4033 | 2239 |
| 5 | CAN | 6.165 | 1.229 | 2.222 | 13.421 | 2981 | 1725 |
| 6 | AUS | 6.281 | 1.544 | 2.252 | 13.449 | 2616 | 1482 |
| 7 | SWE | 6.916 | 1.327 | 2.072 | 12.839 | 2075 | 978 |
| 8 | CHN | 4.300 | 1.064 | 2.715 | 12.087 | 1638 | 1487 |
| 9 | NZL | 5.795 | 1.478 | 3.140 | 12.776 | 1077 | 808 |
| 10 | TPE | 7.569 | 1.634 | 3.362 | 12.960 | 903 | 456 |
| 11 | GRE | 4.867 | 0.738 | 3.868 | 9.180 | 851 | 473 |
| 12 | ITA | 7.067 | 2.467 | 2.718 | 14.365 | 782 | 626 |
| 13 | BEL | 6.236 | 1.258 | 3.734 | 10.462 | 779 | 519 |
| 14 | PHI | 4.697 | 0.712 | 3.025 | 8.425 | 772 | 401 |
| 15 | DEN | 7.498 | 1.263 | 4.910 | 11.947 | 767 | 471 |
| 16 | POL | 7.328 | 1.764 | 3.468 | 13.029 | 744 | 590 |
| 17 | NOR | 6.832 | 1.707 | 3.216 | 11.868 | 692 | 411 |
| 18 | NED | 7.798 | 1.020 | 5.060 | 12.100 | 653 | 408 |
| 19 | FRA | 7.385 | 2.357 | 3.338 | 13.549 | 651 | 605 |
| 20 | HUN | 7.087 | 1.875 | 1.277 | 12.219 | 441 | 289 |
| 21 | THA | 4.189 | 0.934 | 2.982 | 8.778 | 403 | 351 |
| 22 | IRL | 7.247 | 1.534 | 4.605 | 11.667 | 339 | 212 |
| 23 | SUI | 8.190 | 1.546 | 3.586 | 12.905 | 330 | 267 |
| 24 | RUS | 7.787 | 1.137 | 5.693 | 13.318 | 321 | 250 |
| 25 | CZE | 7.385 | 1.475 | 3.298 | 11.558 | 302 | 226 |
| 26 | FIN | 8.111 | 1.059 | 6.480 | 11.666 | 287 | 210 |
| 27 | CRO | 4.876 | 1.016 | 3.355 | 8.116 | 272 | 256 |
| 28 | ARG | 5.114 | 0.783 | 4.063 | 8.281 | 253 | 240 |
| 29 | KOR | 6.483 | 0.855 | 4.623 | 9.944 | 233 | 185 |
| 30 | MAS | 4.193 | 0.980 | 3.096 | 7.645 | 195 | 180 |
| 31 | ESP | 7.344 | 2.004 | 3.953 | 13.411 | 184 | 148 |
| 32 | SGP | 5.685 | 0.685 | 4.816 | 7.842 | 130 | 102 |
| 33 | MEX | 5.635 | 1.067 | 4.031 | 9.070 | 117 | 83 |
| 34 | ROU | 9.299 | 1.157 | 6.938 | 12.533 | 116 | 116 |
| 35 | COL | 4.042 | 0.743 | 3.183 | 6.350 | 96 | 77 |
| 36 | AUT | 5.664 | 2.285 | 3.454 | 11.333 | 92 | 92 |
| 37 | VIE | 4.175 | 0.641 | 3.612 | 6.902 | 75 | 75 |
| 38 | LUX | 5.210 | 0.659 | 4.472 | 7.520 | 70 | 60 |
| 39 | IND | 5.978 | 0.895 | 4.189 | 8.587 | 70 | 70 |
| 40 | HKG | 4.410 | 0.762 | 3.768 | 7.421 | 52 | 52 |
| 41 | CHI | 5.226 | 1.182 | 3.416 | 8.013 | 44 | 41 |
| 42 | SRB | 6.768 | 2.075 | 3.716 | 9.808 | 20 | 17 |
| 43 | ISR | 8.523 | 0.929 | 6.870 | 10.287 | 10 | 10 |

Std (standard deviation); min (minimum value); max (maximum value)

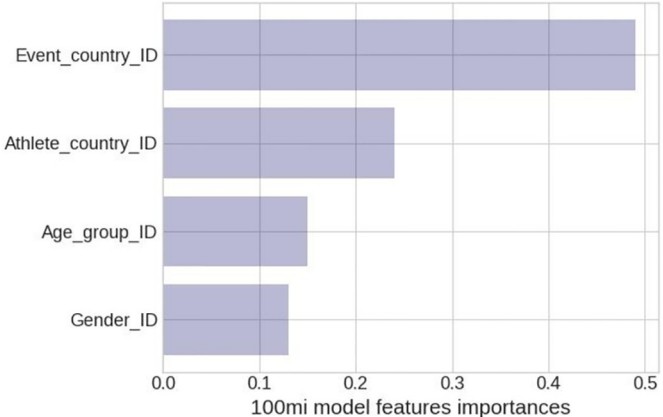

**Fig 2. Optimal model features relative importance.**

boxplots. The difference among male and female predictions is relatively small at about 0.23 km/h (Fig 7). The fastest age group is 25–29 years, but the average speeds of groups 20–24 years and 30–34 years stay close (Fig 8). Participation, however, peaks in the age group 40–44 years. The model replicates predictions that loosely follow the average speed curve in the

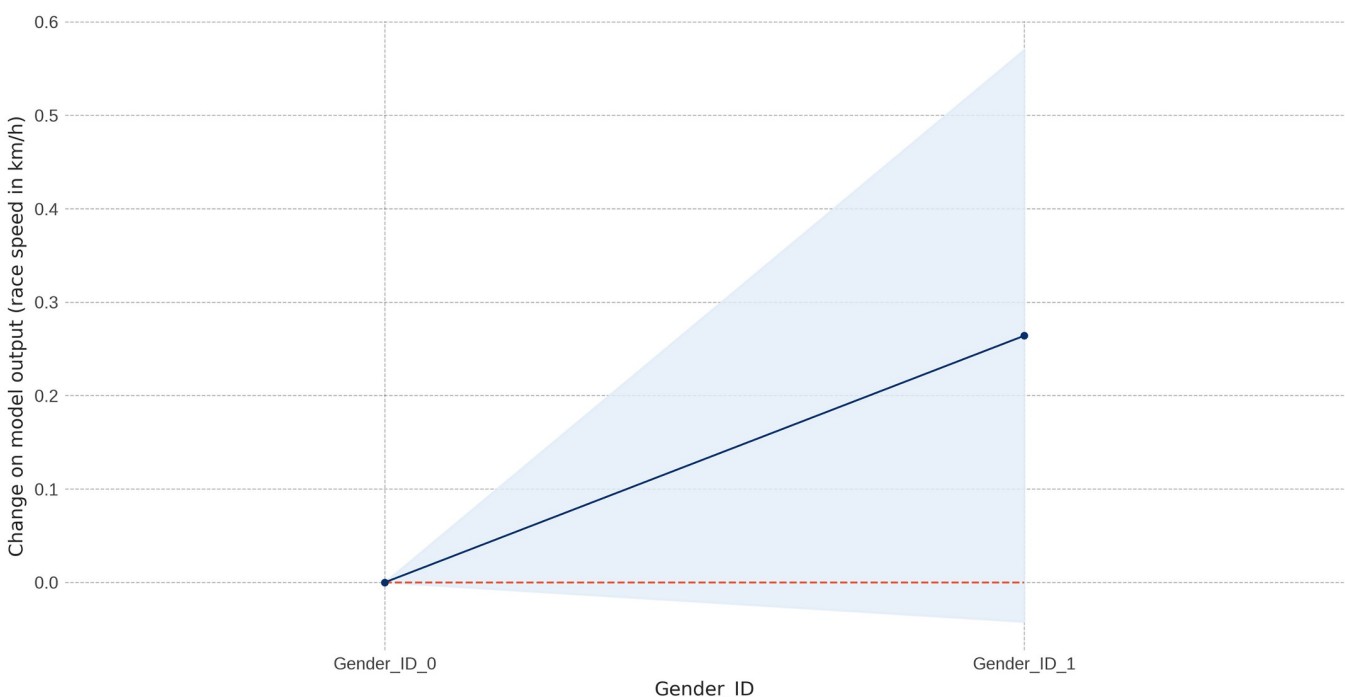

**Fig 3. Partial Dependence Plots (PDP) for gender (ID = female, ID 1 = male).**

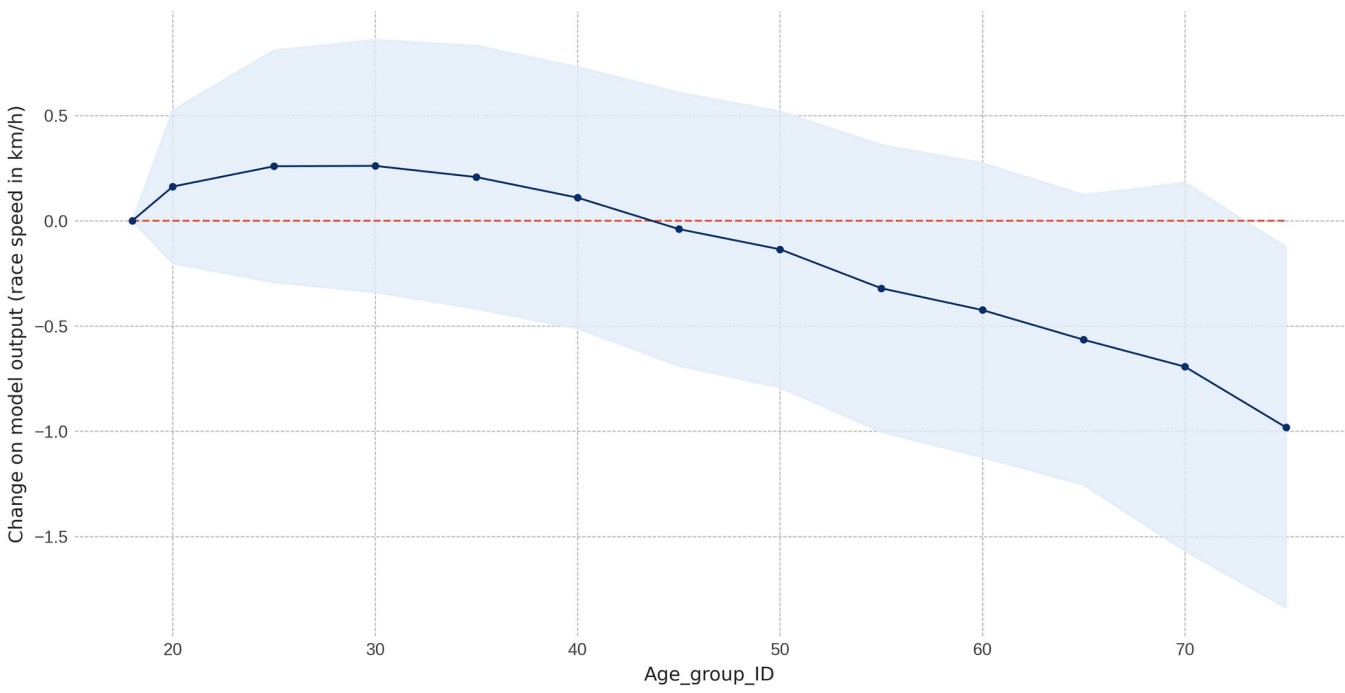

**Fig 4. Partial Dependence Plots (PDP) for age group.**

country-based charts. In terms of participation, the United States leads by far, followed by Great Britain, Canada, South Africa, and Japan, in both athlete and event countries. The fastest athletes come mostly from Eastern European countries (Lithuania, Latvia, Ukraine, Finland, Russia, Hungary, Slovakia) and also Israel, while the slowest athletes come from Asian countries like China, Thailand, Vietnam, or Malaysia (Fig 9). The fastest 100-mile races are held in Romania, Israel, Switzerland, Finland, Russia, the Netherlands, France, Denmark, Czechia, and Taiwan (Fig 10).

## Discussion

The present study aimed to determine the country of origin of the fastest 100-mile runners and the countries hosting the fastest 100-mile race courses using an XG Boost regression model. We found that the event location (*i.e.* the country where the race is held) was the most important predictor for a fast 100-mile race time where the fastest race courses are offered in Romania, Israel, Switzerland, Finland, Russia, the Netherlands, France, Denmark, Czechia, and Taiwan. Regarding the first aim, the fastest athletes come mostly from Eastern European countries (*i.e.*, Lithuania, Latvia, Ukraine, Finland, Russia, Hungary, and Slovakia).

### The fastest race courses

The first important finding was that the country of the event was the most important feature concerning the XG Boost model's predictive power. The countries with the fastest 100-mile

PDP for feature **Athlete_country_ID**

Number of unique grid points: 68

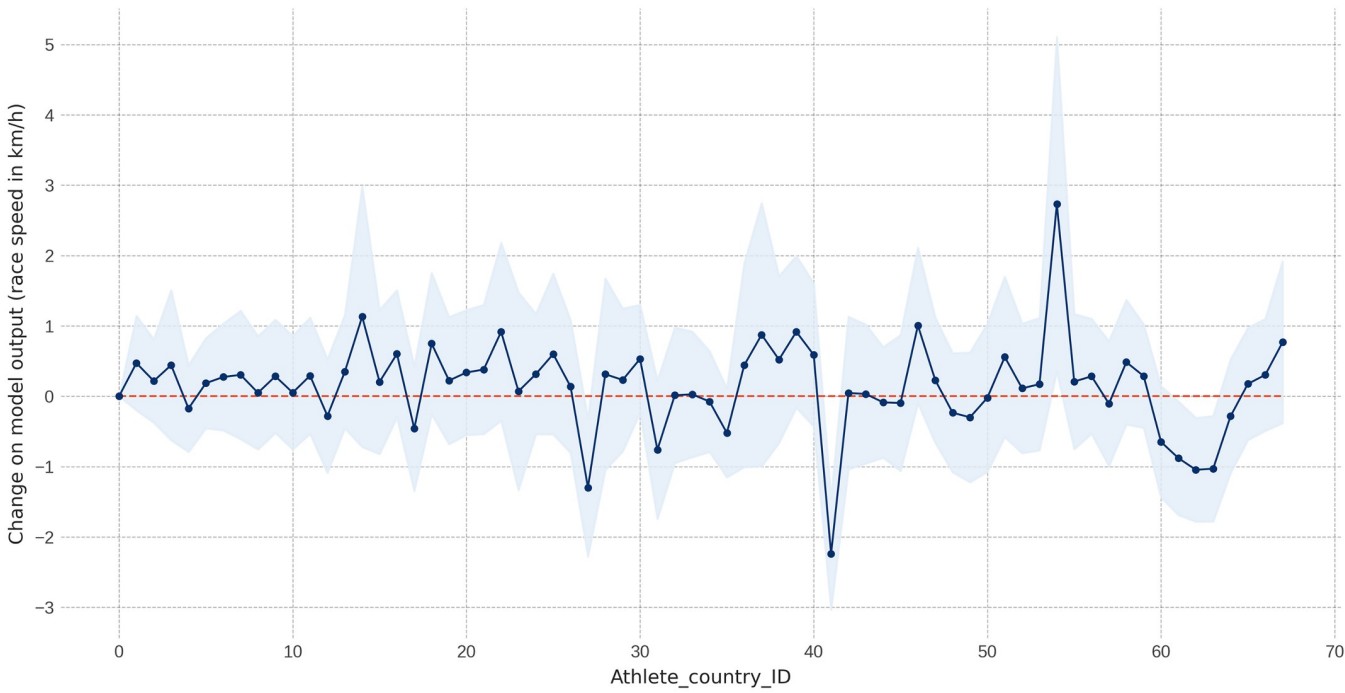

**Fig 5. Partial Dependence Plots (PDP) for the athlete´s country of origin.**

events were Romania, Israel, Switzerland, Finland, Russia, the Netherlands, France, Denmark, Czechia, and Taiwan. Therefore, we could confirm our hypothesis only for Europe, not for Africa and/or America. Common to these races or race courses was the fact they were road-based flat courses on small loops. In some instances, the races recorded the 100-mile split times in a longer or longer race, such as a 24-hour race. In other instances, the races were held as indoor races. In a few instances, the races were held as Championships, such as European or World Championships. In more detail, in Romania, the 'IAU 24 h European Championship' was held in Timisoara in 2018, where the 100-mile split times were taken. The race is a road-based ultra-marathon held on a 1,236 m long asphalt loop (http://s24h.ro/). Importantly, Aleksandr Sorokin from Lithuania passed the 100 miles in 12:50:26 h:min:s. In Israel, the 'Spartanion 100 Miles Race' has been held since 2020 in Ganei Yehoshua Park, Tel Aviv, on a 1,459 m long circular, fast and clean course (https://spartanion.com/). In Switzerland, the '24 heures de Lausanne' recorded 1981 a 100-mile split time with a time of 12:28:16 h:min:s. Furthermore, the '24-Stundenlauf Aare-Insel Brugg' (www.24stundenlauf.ch) and the 'Self-Transcendence 24h Lauf Basel' (https://ch.srichinmoyraces.org/self-transcendence-1224-stunden-lauf-basel) recorded 100-mile split times. In addition, in 1993, the 2nd 'IAU 24h EC Basel' was held with 100 miles split times. In Finland, the 'Endurance 24 h Ultrarun Espoo' has been held since 2010 and the 100-mile split times were taken. The course is a 390,04 m mondo-surfaced indoor track at Esport Ratiopharm Arena in Tapiola Sports Center, Espoo (https://endurance.fi/e24). Different 100-mile races have been held in Russia, such as the '24h 'Sutki Begom' Moskau'

PDP for feature **Event_country_ID**

Number of unique grid points: 44

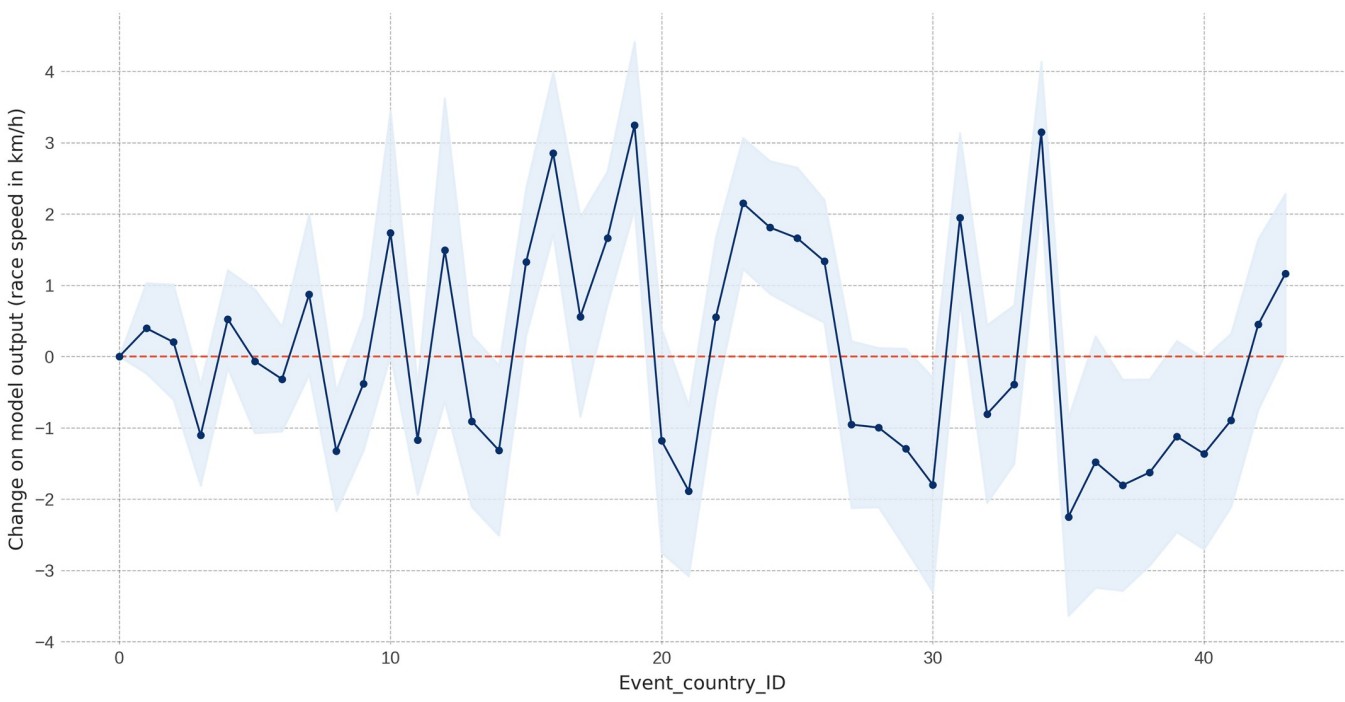

**Fig 6. Partial Dependence Plots (PDP) for country of country where the race was held.**

with a 100-mile split, apart from trail races (Vottovaara Mountain Race and Elton Ultra-Trail). In the Netherlands, the first 100-mile race was held in 1983, with the 'Sint Oedenrode Wande-levenement' held as a walking event. Later, the '24 uurs Apeldoorn' recorded from 1989 to 1997 a 100-mile split time. Furthermore, the 24-hour races '24uur van Steenbergen' and the 'LangsteNachtLoop 24 uurs' recorded 100-mile split times. In France, 100-mile split times were recorded in '48 Heures Pedestre a Montauban', where Yiannis Kouros passed the 100 miles in 1985 in 11:52:40 h:min:s. In 2019, a 100-mile split was recorded in 'IAU 24h WC, 24 heures d'Albi' where Aleksandr Sorokin passed in 13:12:39 h:min:s. The first 100-mile race in Denmark started in 2007 with the 'Mors 100 miles' as a flat road race. In 2009, the '100 Miles —Around the isle of Mors' also started as a flat road race. In Czechia, a 100-mile split time was recorded in the 'Brno Spring 48 Hour Indoor' as an indoor run. Later, split times were recorded in the 'Self-Transcendence Race 24h Kladno' and the 'Běh na 24 hodin Pilsen' as an indoor run. The finding that the country of the race is the most important predictor of perfor-mance might be attributed to topographic characteristics, environmental conditions, and run-ners' preference for specific races to achieve optimal performance. Concerning topographic characteristics, it is observed that most countries with the fastest races share the common fea-ture of flat terrains. In contrast, most of these countries have a continental climate favoring the achievement of fast race. It is also well known that the training process follows the principle of periodization [26, 27], according to which the training is divided into specific phases where the characteristics of exercise (e.g., intensity, volume, recovery and mode) are manipulated to

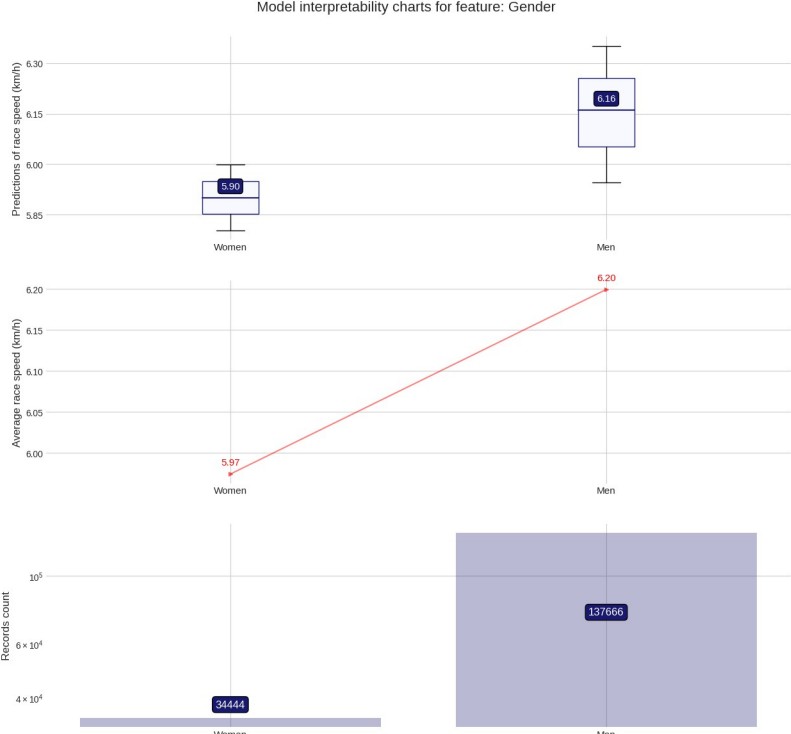

**Fig 7. Prediction distributions and target plots for gender.**

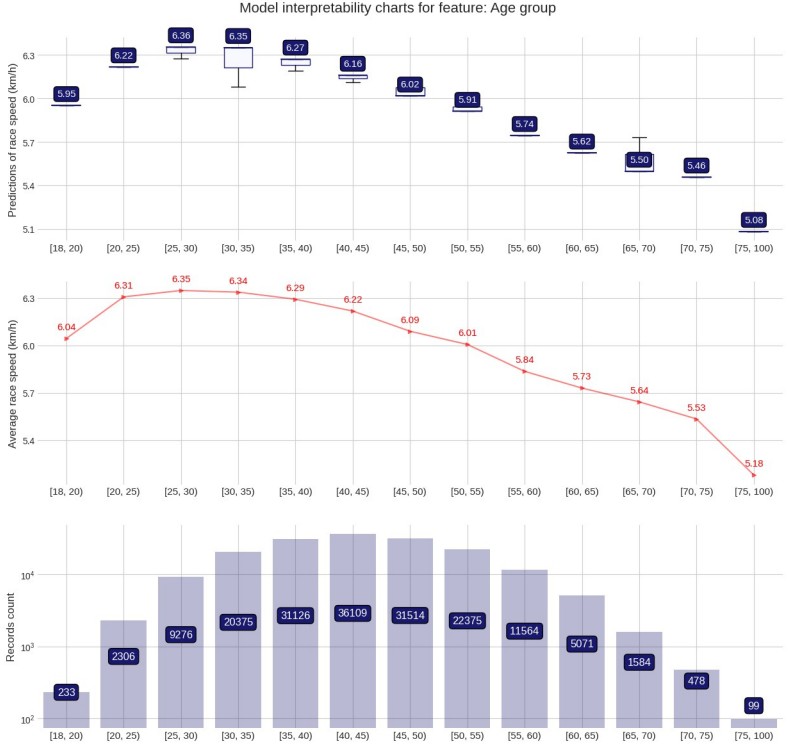

**Fig 8. Prediction distributions and target plots for age group.**

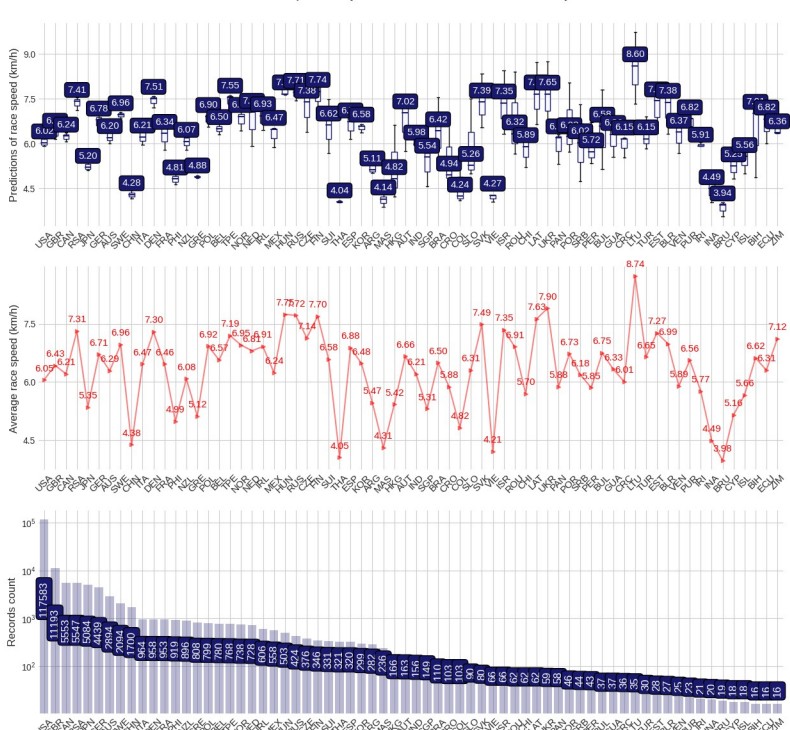

**Fig 9. Prediction distributions and target plots value plots for origin of the athlete.**

peak performance at a certain time. In this context, runners are assumed to participate in a race that fits within their training plan. In addition, a specific race may be selected in terms of reputation (a race can be considered more important than another), where it is already known that other high-level runners intend to participate, and this leads to a sequence of reciprocal cause and effect in which: the fast runners choose fast races to compete, and in turn, the participation of fast runners ensure that a fast race remains fast.

## The fastest runners

In contrast to a recent study reporting that the fastest 100-mile ultra-marathoners were women from Sweden, Hungary and Russia and men from Brazil, Russia and Lithuania [25], we found that runners from Lithuania, Latvia, Ukraine, Finland, Russia, Hungary, and Slovakia obtained the fastest running speeds. In the first instance, we found that 35 runners from Lithuania were among the fastest. Although it might be possible that one or a few runners from the same country could bias the result, the best Lithuanian ultra-marathoner, Aleksandr 'Sania' Sorokin, has finished only four 100-mile races. Still, with the world record 100 miles on the track in 2021 in the 'Centurion Running Track 100 Mile' in the United Kingdom and the 100 miles on road in the 'Sparanion Race' in 2022 in Israel (www.irunfar.com/aleksandr-sorokin-150-kilometer-100-mile-and-12-hour-world-record-holder-interview). Therefore, 31 race records must be from other fast Lithuanian ultra-marathoners. It should be highlighted that the fastest runners in the present study originated from countries that shared geographical, cultural, and socio-economical characteristics. Furthermore, a recent review reported a dominance of Russian athletes in ultra-marathon running and suggested as potential explanations a possible misuse of

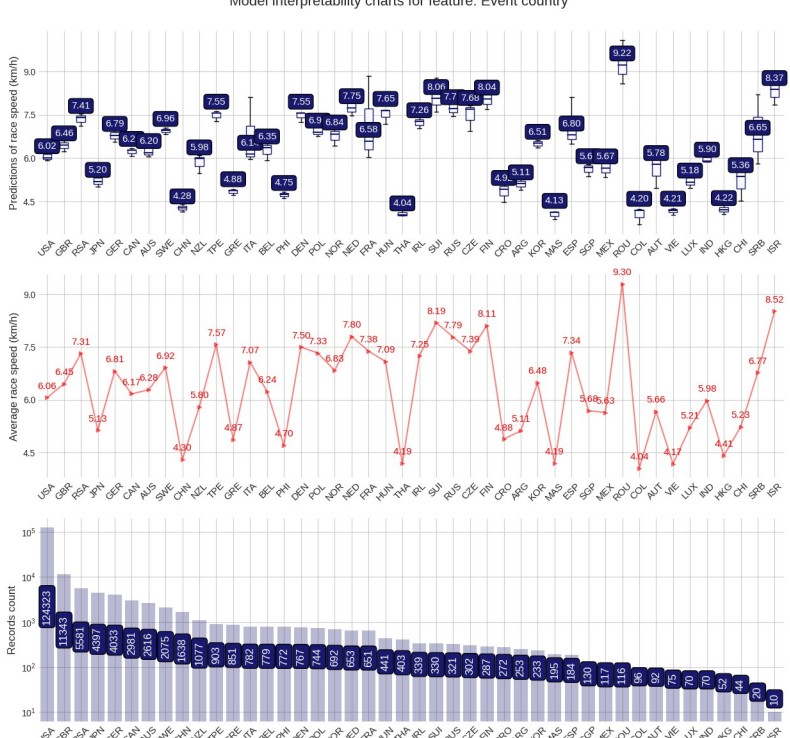

**Fig 10. Prediction distributions and target plots value plots for the country where the events were held.**

performance-enhancing substances, historical, climate-geographical, and psychophysiological (e.g., a combination of genetic and social) factors [28]. Although most 100-mile runners were from the United States, US runners are not among the fastest. In the US, plenty of 100-mile races are held, and most are trail runs (https://runningintheusa.com/classic/list/map/100m). One of the most traditional 100-mile races is the 'Western States 100 Mile Endurance Run' held since 1976 (www.ws100.com/). Another 100-mile race with a long tradition is the 'Old Dominion 100 Mile Endurance Run', which started in 1979 (www.olddominionrun.org/). The greater participation of US runners in the race may shift the average time downwards, which does not necessarily mean that they have lower times than other nationalities. The relatively large number of US-American finishers in this race distance indicated that these runners could be more 'recreational' than those from other countries (who, in turn, could be considered more 'selective') and might partially explain that they were not among the fastest nationalities.

## The age of peak performance

We also found that athletes in the age group 25 (25–29 years) were the fastest in the 100-mile race distance. This age is significantly lower than that found in a study of 35,956 finishes (6,862 women and 29,094 men) in 100-mile ultramarathons between 1998 and 2011. The annual top ten fastest runners had an average age of ~39 and ~37 years for women and men [29]. The difference to the present results might be that the present study considered all athletes, whereas the existing study was restricted to the annual ten fastest. Furthermore, the relatively young age of the fastest finishers in our study might be explained in terms of 'selectiveness' variation by age group. The number of finishers in the 25–29 age group is three to four times less than

that in the age groups 35–39, 40–44 and 45–49, suggesting that the athletes in the former one might be considered as more 'selected' compared to the more 'recreational' athletes of the latter groups. In another way, these are interesting findings, indicating that young runners, when well-trained, can perform well in ultramarathon events.

## Limitations

Although this study uses a very large data set and highly sophisticated analyses, we must acknowledge some limitations. We found that the fastest running speeds were obtained by runners from Lithuania, Latvia, Ukraine, Finland, Russia, Hungary, and Slovakia—countries with partially low numbers of runners. Since we did not account for repeated measures, one or two outstanding athletes from these countries could be responsible for the country's performance. However, as described by Aleksandr 'Sania' Sorokin, only one athlete cannot achieve all the best race results for one country. Aspects such as training, previous experience [30], motivation [31], drafting [32], pre-race nutrition [33], and environmental conditions [34] could not be considered. We must also be aware that these race courses might not all have been exactly measured, so some very fast race courses might not have the full length of 100 miles (161 km). Another limitation is associated with the available information. With only four predictors, the model could only be very general. More realistic models could be built by collecting additional runner-specific data and mixing it with the available data.

## Conclusion

In summary, the event location (*i.e.* the country where the race is held) is the most important predictor for a fast 100-mile race time, according to our XG Boost regression model. The fastest race courses occurred in Romania, Israel, Switzerland, Finland, Russia, the Netherlands, France, Denmark, Czechia, and Taiwan. Common to these races or race courses is the fact they are held on a road-based flat course on small loops. In some instances, the races took the 100-mile split times in a longer race, such as a 24-hour race or longer. In other instances, the races were held as indoor races. In a few instances, the races were held as European or World Championships. Athletes and coaches can use these findings for their race preparation to find the most appropriate race course for a fast 100-mile race time. For example, running a 24-hour race (often flat and circular) might be better to try to break 100-mile personal best time, thus combining two "races" in one, than running some challenging 100-mile race.

## Supporting information

**S1 Data.**
(XLSX)

**S2 Data.**
(XLSX)

## Author Contributions

**Conceptualization:** Beat Knechtle.

**Data curation:** Elias Villiger.

**Formal analysis:** David Valero.

**Writing – review & editing:** Katja Weiss, Pantelis T. Nikolaidis, Marilia Santos Andrade, Volker Scheer, Ivan Cuk, Robert Gajda, Mabliny Thuany.

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
