## [Decision Letter · Decision Letter 0]

12 Feb 2024

PONE-D-23-43701Using Machine Learning to Determine the Nationalities of the Fastest 100-Mile Ultra-Marathoners and Identify Top Racing EventsPLOS ONE

Dear Dr. Knechtle,

Thank you for submitting your manuscript to PLOS ONE. After careful consideration, we feel that it has merit but does not fully meet PLOS ONE’s publication criteria as it currently stands. Therefore, we invite you to submit a revised version of the manuscript that addresses the points raised during the review process.

We look forward to receiving your revised manuscript.

Kind regards,

Stevo Popovic, Ph.D.

Academic Editor

PLOS ONE

3. In the online submission form you indicate that your data is not available for proprietary reasons and have provided a contact point for accessing this data. Please note that your current contact point is a co-author on this manuscript. According to our Data Policy, the contact point must not be an author on the manuscript and must be an institutional contact, ideally not an individual. Please revise your data statement to a non-author institutional point of contact, such as a data access or ethics committee, and send this to us via return email. Please also include contact information for the third-party organization, and please include the full citation of where the data can be found.

Additional Editor Comments:

I am so please to inform you I have collected two reviews and we are ready to go ahead with the evaluation process. It is your turn now to read the reviews and carefully revise the manuscript according to the requirements of the reviewers. I would appreciate if you prepare the comments with some arguments and adequate justifications for both of them.

Reviewers' comments:

Reviewer's Responses to Questions

**Comments to the Author**

1. Is the manuscript technically sound, and do the data support the conclusions?

Reviewer #1: Yes

Reviewer #2: Partly

2. Has the statistical analysis been performed appropriately and rigorously? 

Reviewer #1: Yes

Reviewer #2: N/A

3. Have the authors made all data underlying the findings in their manuscript fully available?

Reviewer #1: Yes

Reviewer #2: Yes

4. Is the manuscript presented in an intelligible fashion and written in standard English?

Reviewer #1: Yes

Reviewer #2: Yes

5. Review Comments to the Author

Reviewer #1: Manuscript Number: PONE-D-23-43701

Manuscript Title: Using Machine Learning to Determine the Nationalities of the Fastest 100-Mile Ultra-Marathoners and Identify Top Racing Events

The manuscript is interesting and well written with several strengths.

I just have a few minor comments to make.

- Overall, the different sections of the manuscript are well written.

- Line 50: delete “(ML)”.

- Line 60: Please, change “offered” to “occured”.

- Lines 79 – 80: Missed reference.

- Lines 103 – 105: Change “In this study, therefore, we sought to determine the country of origin of the fastest runners and the location of the fastest race courses.” to “In this context, we undertook this research to determine the country of origin of the fastest runners and the location of the fastest race courses.”

- Line 142: Change “ML” to “machine learning (ML)”

- Line 165: What is “MAE”

- Line 257 - 269: The first paragraph of the discussion is too large, I would suggest creating a small paragraph.

- Line 258 – 260: Replace “Based on a recent study, we assumed that the fastest athletes would be found to originate from Sweden, Hungary, Russia, Brazil, or Lithuania” to “Based on a recent study, we assumed that the fastest athletes would come from Sweden, Hungary, Russia, Brazil, or Lithuania.”

- Line 374: delete “had”.

- Line 376: delete “Better”.

- Line 384: Change “offered” to “occurred”.

- The limitations are properly discussed.

- Figures 3, 4, 5, and 6: change “PDP” to “partial dependence plots (PDP)”.

Reviewer #2: The work represents an interesting attempt to apply the XGBoost algorithm to determine the country of origin of the fastest runners and to identify the fastest races. The input characteristics under consideration are: athletes nationality, age group, gender, as well as the event country, while the output is the estimation of speed. The mentioned application has several limitations, the most important of which is the selection of input characteristics, i.e., the ability to obtain a good prediction from them. Another limitation is the dataset, which contains data from different types of races (championships, recreational races), undoubtedly resulting in a biased estimator. Additionally, for certain countries, there is a very small number of samples, and samples are with a high standard deviation. All of this does not lead to a high-quality estimator. However, my recommendation is for the work to be published, considering that the authors are aware of all the shortcomings of the estimator they have made, they have adequately addressed them in the paper and addressed potential risks, so the work can be a good guide for future research in this area.

I suggest incorporating an analysis of the median as it often provides a more robust measure than the mean, particularly when dealing with datasets containing significant deviations in extreme values.

6. PLOS authors have the option to publish the peer review history of their article (what does this mean?). If published, this will include your full peer review and any attached files.

Reviewer #1: **Yes: **Nejmeddine Ouerghi

Reviewer #2: No

---

## [Author Response · Author response to Decision Letter 0]

31 Mar 2024

Reviewers' comments:

Reviewer's Responses to Questions

Comments to the Author

1. Is the manuscript technically sound, and do the data support the conclusions?

Reviewer #1: Yes

Reviewer #2: Partly

2. Has the statistical analysis been performed appropriately and rigorously? 

Reviewer #1: Yes

Reviewer #2: N/A

3. Have the authors made all data underlying the findings in their manuscript fully available?

Reviewer #1: Yes

Reviewer #2: Yes

4. Is the manuscript presented in an intelligible fashion and written in standard English?

Reviewer #1: Yes

Reviewer #2: Yes

5. Review Comments to the Author

Reviewer #1: Manuscript Number: PONE-D-23-43701

Manuscript Title: Using Machine Learning to Determine the Nationalities of the Fastest 100-Mile Ultra-Marathoners and Identify Top Racing Events

The manuscript is interesting and well written with several strengths.

I just have a few minor comments to make.

- Overall, the different sections of the manuscript are well written.

- Line 50: delete “(ML)”.

Answer: We agree with the expert reviewer and deleted as suggested.

- Line 60: Please, change “offered” to “occured”.

Answer: We agree with the expert reviewer and changed as suggested

- Lines 79 – 80: Missed reference.

Answer: We agree with the expert reviewer and added as suggested

- Lines 103 – 105: Change “In this study, therefore, we sought to determine the country of origin of the fastest runners and the location of the fastest race courses.” to “In this context, we undertook this research to determine the country of origin of the fastest runners and the location of the fastest race courses.”

Answer: We agree with the expert reviewer and changed as suggested

- Line 142: Change “ML” to “machine learning (ML)”

Answer: We agree with the expert reviewer and changed as suggested

- Line 165: What is “MAE”

Answer: We agree with the expert reviewer and changed to MAE (Mean Absolute Error)

- Line 257 - 269: The first paragraph of the discussion is too large, I would suggest creating a small paragraph.

Answer: We agree with the expert reviewer and changed as suggested to: The present study aimed to determine the country of origin of the fastest 100-mile runners and the countries hosting the fastest 100-mile race courses using an XG Boost regression model. We found that the event location (i.e. the country where the race is held) was the most important predictor for a fast 100-mile race time where the fastest race courses are offered in Romania, Israel, Switzerland, Finland, Russia, the Netherlands, France, Denmark, Czechia, and Taiwan. Regarding the first aim, the fastest athletes come mostly from Eastern European countries (i.e., Lithuania, Latvia, Ukraine, Finland, Russia, Hungary, and Slovakia).

- Line 258 – 260: Replace “Based on a recent study, we assumed that the fastest athletes would be found to originate from Sweden, Hungary, Russia, Brazil, or Lithuania” to “Based on a recent study, we assumed that the fastest athletes would come from Sweden, Hungary, Russia, Brazil, or Lithuania.”

Answer: since we had to reduce that section this sentence was deleted

- Line 374: delete “had”.

Answer: We agree with the expert reviewer and deleted as suggested.

- Line 376: delete “Better”.

Answer: We agree with the expert reviewer and deleted as suggested.

- Line 384: Change “offered” to “occurred”.

Answer: We agree with the expert reviewer and changed as suggested.

- The limitations are properly discussed.

Answer: no changes are required

- Figures 3, 4, 5, and 6: change “PDP” to “partial dependence plots (PDP)”.

Answer: We agree with the expert reviewer and changed as suggested.

Reviewer #2: The work represents an interesting attempt to apply the XGBoost algorithm to determine the country of origin of the fastest runners and to identify the fastest races. The input characteristics under consideration are: athletes nationality, age group, gender, as well as the event country, while the output is the estimation of speed. The mentioned application has several limitations, the most important of which is the selection of input characteristics, i.e., the ability to obtain a good prediction from them. Another limitation is the dataset, which contains data from different types of races (championships, recreational races), undoubtedly resulting in a biased estimator. Additionally, for certain countries, there is a very small number of samples, and samples are with a high standard deviation. All of this does not lead to a high-quality estimator. However, my recommendation is for the work to be published, considering that the authors are aware of all the shortcomings of the estimator they have made, they have adequately addressed them in the paper and addressed potential risks, so the work can be a good guide for future research in this area.

I suggest incorporating an analysis of the median as it often provides a more robust measure than the mean, particularly when dealing with datasets containing significant deviations in extreme values.

Answer: We recognize the limitations of this analysis, but could only use the data available. As a predictive model, there is no question about our XGB model ability (R2=0.23 in-sample tested). So the main strength of our study resides in using the model interpretability tools (reverse engineering) such as the PDP charts and the prediction distribution charts, to understand the 23% of explained output variability. Interestingly, but not as a surprise, the XGB model mostly learnt the descriptive statistical structure of the dataset.

On the comment about the median, I would like to emphasize that there is indeed a mix of mean and median values in the charts. So I´ll explain here in detail, so there are no doubts about this.

The country ranking tables (sorted by number of race records) are created simply by aggregating race records and then calculating the different stats. So they show the basic descriptive statistics (mean, std, min and max and the counts).

The PDP charts are produced after the model has been trained and represent the relative value of the model output for a predictor. The curve highlights the mean values.

Finally, the called “model interpretability charts” show group average values (means) in the middle chart (red curve) and median values in the top chart (boxplot with blue labels). Actually they are nearly the same.

---

## [Decision Letter · Decision Letter 1]

6 May 2024

Using Machine Learning to Determine the Nationalities of the Fastest 100-Mile Ultra-Marathoners and Identify Top Racing Events

PONE-D-23-43701R1

Dear Dr. Knechtle,

We’re pleased to inform you that your manuscript has been judged scientifically suitable for publication and will be formally accepted for publication once it meets all outstanding technical requirements.

Kind regards,

Stevo Popovic, Ph.D.

Academic Editor

PLOS ONE

Additional Editor Comments (optional):

Reviewers' comments:

Reviewer's Responses to Questions

**Comments to the Author**

1. If the authors have adequately addressed your comments raised in a previous round of review and you feel that this manuscript is now acceptable for publication, you may indicate that here to bypass the “Comments to the Author” section, enter your conflict of interest statement in the “Confidential to Editor” section, and submit your "Accept" recommendation.

Reviewer #1: All comments have been addressed

Reviewer #2: All comments have been addressed

2. Is the manuscript technically sound, and do the data support the conclusions?

Reviewer #1: Yes

Reviewer #2: Yes

3. Has the statistical analysis been performed appropriately and rigorously? 

Reviewer #1: Yes

Reviewer #2: (No Response)

4. Have the authors made all data underlying the findings in their manuscript fully available?

Reviewer #1: (No Response)

Reviewer #2: (No Response)

5. Is the manuscript presented in an intelligible fashion and written in standard English?

Reviewer #1: Yes

Reviewer #2: (No Response)

6. Review Comments to the Author

Reviewer #1: Manuscript Number: PONE-D-23-43701R1

Manuscript Title: Using Machine Learning to Determine the Nationalities of the Fastest 100-Mile Ultra-Marathoners and Identify Top Racing Events

The manuscript is interesting and nicely written.

The authors have been well corrected and modified the manuscript according to my comments.

I recommend to accept the manuscript for publication.

Reviewer #2: (No Response)

7. PLOS authors have the option to publish the peer review history of their article (what does this mean?). If published, this will include your full peer review and any attached files.

Reviewer #1: **Yes: **Nejmeddine Ouerghi

Reviewer #2: No

---

## [Editor Report · Acceptance letter]

20 May 2024

PONE-D-23-43701R1 

PLOS ONE

Dear Dr. Knechtle, 

I'm pleased to inform you that your manuscript has been deemed suitable for publication in PLOS ONE. Congratulations! Your manuscript is now being handed over to our production team.

Kind regards, 

on behalf of

Professor Stevo Popovic 

Academic Editor

PLOS ONE